# Testing Different Message Styles about Unnecessary Antibiotics Using an Online Platform

**DOI:** 10.3390/antibiotics13070657

**Published:** 2024-07-16

**Authors:** Säde Stenlund, Kirstin C. Appelt, Matthew B. Ruby, Nick Smith, Hannah Lishman, David M. Patrick

**Affiliations:** 1BC Centre for Disease Control, Vancouver, BC V5Z 4R4, Canadadavid.patrick@ubc.ca (D.M.P.); 2School of Population and Public Health, University of British Columbia, Vancouver, BC V6T 1Z3, Canada; 3Department of Public Health, University of Turku, 20014 Turku, Finland; 4Sauder School of Business, University of British Columbia, Vancouver, BC V6T 1Z2, Canada; 5School of Psychology and Public Health, La Trobe University, Bundoora, VIC 3086, Australia

**Keywords:** antibiotic resistance, antibiotic messaging, antibiotic prescribing, antimicrobial stewardship, health communication

## Abstract

Patients’ expectations are a major contributor to the unnecessary prescribing of antibiotics, yet limited research has examined how physicians can calibrate these expectations. The studies we conducted tested how varying messages could impact patients’ expectations for antibiotics and their experience of medical appointments. All the participants read a short scenario about an appointment for mild sinusitis symptoms, with the patient’s expectation of antibiotics. In Study 1, the participants (*n* = 1069) were randomly assigned to read a positively framed, neutral, or negatively framed message regarding unnecessary antibiotics. In Study 2, the participants (*n* = 1073) read a message emphasizing either the societal or personal harms of unnecessary antibiotics, or a message without additional rationale. None of our pre-registered hypotheses were supported, but our exploratory analyses indicated that the societal message increased concern about antibiotic resistance. The participants who were more concerned about resistance were less likely to ask for antibiotics, more satisfied when the physician did not prescribe them, and more likely to recommend the physician to a friend. Discussing the consequences of the different courses of action did not appear to negatively impact physician–patient rapport. These studies demonstrate an inexpensive method with which to pre-test various messages about antibiotic consumption, and suggest that such messages are not negatively received by patients.

## 1. Introduction

Antibiotic resistance is one of the leading global health threats we face [1]. Reducing the use of antibiotics that lack a clear medical indication is a key strategy to address antimicrobial resistance [2]. During the past decades, antimicrobial stewardship efforts have changed guidelines, to nudge physicians towards more restricted antimicrobial prescribing [3]. However, changing prescribing behavior takes time [4], and non-medically necessary prescriptions remain a significant proportion of the prescribed antibiotics, with an estimated rate as high as one-third in the US [5] and about 15% in Canada [6]. In cases where the guidance is changing and physicians’ antibiotic-prescribing practices vary, patients’ expectations may vary as well, with many patients still expecting to be prescribed antibiotics. Patients might not acknowledge the risks associated with unnecessary prescribing, due to incomplete knowledge about antimicrobial resistance [7,8]. Furthermore, physicians have reported that patients’ expectations are a major contributor to unnecessary prescribing [9,10], even though these perceived expectations do not always correlate with patients actually expecting to be prescribed antibiotics [11]. Despite the importance of patients’ expectations, research is limited on how physicians can work with patients to calibrate their expectations. The present studies employ a novel method for the preliminary testing of various messaging strategies that a physician could employ to reduce patients’ expectations for antibiotics.

Past research on the factors affecting whether antibiotics are prescribed has mostly relied on observational data (e.g., [11]). An alternative to this approach is to conduct experiments to causally test which factors lead to changes in expectations. A cost-effective method is to use online survey platforms, such as Qualtrics, and online participant panels, such as MTurk and Prolific. This method has been utilized previously for studying whether providing additional information about the viral nature of an infection and participants’ trust in their physician affects whether they expect to be prescribed antibiotics [12]. To the best of our knowledge, this method has not been used to test the impact of different messages about antibiotic resistance and the consequences of unnecessary prescribing.

The verbal messages a physician shares with a patient are one of the more powerful tools with which to change patient perceptions and affect their behavior [13]. Behavioral science provides insights into how different messages yield different responses. For example, research on goal framing has found that describing a decision in terms of the negative consequences of choosing one option is more effective than describing the same decision in terms of the positive consequences of choosing the other option [14]. In the context of antibiotic prescribing, this might involve focusing on the negative consequences of unnecessary antibiotics rather than the positive consequences of avoiding antibiotics. In addition, research on motivation suggests that appealing to a person’s selfish motives might have a different effect than emphasizing altruistic reasons [15]. Translating these findings to the context of antibiotic prescribing, antimicrobial stewardship efforts often focus on the societal threats of antimicrobial resistance to tap into altruistic motivation. These same messages could instead focus on the adverse personal side effects to provide a selfish argument to avoid unnecessary antibiotics [16,17]. Thus, the framing and motivation of physicians’ messaging can both be manipulated. The present studies explore which combination of these messages would be most impactful for reducing patient expectations for antibiotics.

The present studies focus on sinusitis. The treatment of sinusitis has been shifting toward a more symptomatic approach, especially for milder cases [6,18]. However, both physicians’ antibiotic-prescribing practices and patient expectations still vary [5]. The recommendations for mild cases of sinusitis are to avoid prescribing antibiotics, but patients may still expect to be prescribed antibiotics. Therefore, sinusitis provides a suitable context for testing various messaging strategies to reduce unnecessary antibiotic prescribing.

The aim of these studies was to test how different messages warning against unnecessary prescribing would impact people’s expectation for antibiotics, as well as their satisfaction with the appointment. In our pre-registration we hypothesized that the negative message would lead to a greater reduction in demand for antibiotics. We also hypothesized that the message about personal benefits (i.e., selfish motives) would be more effective than the societal one.

## 2. Study 1

In Study 1, we tested the impact of message framing on patient expectations. Ethical approval for this study was obtained from the Institutional Review Board of the University of British Columbia (H23-02242). The participants provided their informed consent when entering this study.

### 2.1. Materials and Methods

#### 2.1.1. Study Population

In the absence of estimations of the meaningful effect size in the published literature, we estimated our required sample as follows: Through clinical experience, we expected 50% of patients to still ask for antibiotics, and considered that a decrease of 10% in these requests would be meaningful. To detect such a difference in three conditions with 0.80 power and *α* = 0.05 would require 246 participants per condition. However, if we expected the underlying prevalence of the expectation to be different, and the consequent change to be from 80 to 70% or 30 to 20%, we would need 358 participants per condition.

We recruited participants from Prolific, an online platform commonly used to recruit research participants in the behavioral sciences; for an overview of this platform and its reliability, see [19]. We recruited 1075 participants, who were currently residing in the US or Canada and were aged 18 years or older. The sample was balanced for gender. Six participants failed the attention check and were excluded, resulting in a final sample of 1069 participants.

The participants’ characteristics are described in Table 1. The largest proportion of the participants had a post-secondary degree (46%, *n* = 496) and most of the participants rated their health as good, very good, or excellent (87%, *n* = 931). The sample was balanced for gender.

#### 2.1.2. Procedure

This study was approved by the Research Ethics Board of the University of British Columbia and pre-registered with AsPredicted (https://aspredicted.org/4hr42.pdf (accessed on 1 March 2024)). After reading a short description of this study and indicating their consent to participate, the participants read a scenario about experiencing symptoms of sinusitis and booking a physician appointment to obtain an antibiotic prescription, as in previous cases (see Section 2.1.3 below). In the scenario, the doctor advises against antibiotics for mild sinusitis and instead recommends symptom treatment.

The participants were randomized into three conditions which provided different rationales for avoiding antibiotics in this case. They read a negative framing that emphasized the harmful effects of taking unnecessary antibiotics, a positive framing that emphasized the protective effects of not taking unnecessary antibiotics, or a control condition that did not include an additional rationale.

After reading the scenario, the participants answered a set of questions about the situation and their basic demographics, including their age, perceived health, gender, and education. An attention check was included that asked participants to “Please select “disagree” on this item”. The participants received USD 1.10 for taking part in this study.

#### 2.1.3. Scenario

All the participants read the base scenario. As indicated below, the third paragraph varied by condition:

Imagine that you’ve had cold symptoms for a week. Today you woke up feeling worse. Your nose is congested and you feel mucus dripping down your throat. You feel pressure and pain around your nose. You make an appointment to see a doctor today. During the appointment, you plan to ask for a prescription for antibiotics because the cold is getting worse rather than better.

The doctor examines you and states that you have sinusitis. However, he says that he does not recommend antibiotics to treat sinusitis because research has shown that antibiotics do not shorten the duration of symptoms.

Control condition: {No additional text.}

Negative condition: {Furthermore, taking antibiotics harms your gut microbiome. When you take antibiotics, they can kill the good bacteria in your gut. This, in turn, makes your gut more vulnerable to pathogens and has been linked to worse health outcomes for many different conditions.}

Positive condition: {Furthermore, avoiding antibiotics protects your gut microbiome. When you avoid antibiotics, this can save the good bacteria in your gut. This, in turn, makes your gut more resistant to pathogens and has been linked to better health outcomes for many different conditions.}

Therefore, he does not advise taking antibiotics. Instead, he recommends treating the symptoms by rinsing your nose twice a day with salt water. Research has shown that this procedure can reduce the severity and duration of symptoms.

#### 2.1.4. Measures

As stated in our pre-registration, our goal was to test whether our manipulation influenced the three main outcome variables. The participants answered the following questions on 1–5 Likert-type scales: (1) How likely are you to still ask for an antibiotic prescription? (1—very unlikely–5—very likely); (2) If you left the appointment without a prescription, how satisfied would you feel with the appointment overall? (1—very dissatisfied–5—very satisfied); and (3) How likely would you be to recommend this doctor to a friend? (1—very unlikely–5—very likely). Additional secondary outcome variables included the following items: If you left the appointment without a prescription, how likely would you be to seek out a second opinion from another doctor? (1—very unlikely–5—very likely); How likely is it that you would return to this doctor the next time you are ill? (1—very unlikely–5—very likely); and How concerned are you about antimicrobial resistance? (1—not at all concerned–5—very concerned). Participants were also asked whether they had previously had sinusitis and received antibiotics for it, and whether they had been previously advised by a doctor that antibiotics are not the best treatment option.

#### 2.1.5. Statistical Analyses

As stated in our pre-registration, we conducted a three-way ANOVA for each outcome variable. Using the same method, the exploratory analyses tested whether the manipulation had any effect on the participants’ concerns about resistance, likelihood of seeking a second opinion, and likelihood of returning to the physician. The participants’ concerns about resistance were correlated with the primary and secondary outcomes. Linear models were used to test how personal characteristics might affect concerns about resistance.

### 2.2. Results

In contrast to our hypotheses, we did not observe any significant differences between the positive, negative, or neutral conditions on any of our outcome variables (*p* = 0.18–0.79, Table 2). Although the participants in the negative frame were slightly less likely to continue to request antibiotics, were more satisfied with their appointment, and were more likely to recommend the doctor to a friend, none of these differences were significant.

In the exploratory analysis, we found that the manipulation did not have a significant effect on how concerned the participants were about antimicrobial resistance (*F* = 1.16, *p* = 0.31), but did on whether the participants would be more likely to seek a second opinion (*F* = 4.34, *p* = 0.01; neutral vs. negative, *p* = 0.009). The participants who were more concerned about antibiotic resistance were less likely to ask for antibiotics (*r* = −0.32, *p* < 0.001) or to seek a second opinion (*r* = −0.26, *p* < 0.001); they were more likely to be satisfied with the appointment (*r* = 0.28, *p* < 0.001), return to the same doctor (*r* = 0.29, *p* < 0.001), and recommend the doctor (*r* = 0.30, *p* < 0.001).

Overall, the participants were somewhat concerned about antimicrobial resistance (*M* = 3.20; on a scale from 1—not at all concerned to 5—very concerned). A third of the participants (*n* = 365, 34%) had previously had sinusitis and, of these participants, almost half (45%) reported previously receiving antibiotics for their sinusitis. Of the patients who previously received antibiotics, 29% had been advised that antibiotics are not the best treatment, compared to 63% who had not heard this. Of the patients who had not previously received antibiotics for their sinusitis, 46% had heard this message, whereas 41% had not. A higher concern about resistance was observed among the participants who reported previously hearing that antibiotics are not the best treatment (*b* = −0.27, *p* < 0.001), and those with higher education (*b* = 0.018, *p* < 0.001). No significant differences in concerns about resistance were observed based on age, perceived health, or gender.

## 3. Study 2

In Study 2, we tested the impact of personal vs. societal motivation. As stated in our pre-registration, we used the negative frame from Study 1 as the basis of the scenario in Study 2. Ethical approval for this study was obtained from the Institutional Review Board of the University of British Columbia (H23-02242). The participants provided their informed consent when entering this study.

### 3.1. Materials and Methods

In Study 2, we performed identical power calculations, recruitment of participants, and procedures as those in Study 1, with the following exceptions: (1) no participants who had taken part in Study 1 could participate in Study 2, and (2) the scenarios compared a personal vs. societal message.

For Study 2, we recruited 1075 participants. One response was discarded by the Prolific platform, and one additional participant failed the attention check, resulting in a total of 1073 participants. The participants’ characteristics are presented in Table 3. The largest proportion of the participants had a post-secondary degree (45%, *n* = 485), and most of the participants rated their health as good, very good, or excellent (86%, *n* = 919). The sample was balanced for gender.

#### 3.1.1. Scenario

The scenario was identical to that of Study 1, except for the additional rationale. Because the conditions in Study 1 did not differ significantly, the negative condition was chosen as the base message, in line with our pre-registration. The participants were, therefore, randomized into groups, reading either a negative framing of the personal threats of antibiotics (same as the negative condition in Study 1), a negative framing of the societal threats of antibiotic resistance, or no additional argument.

Control condition: {No additional text.}

Personal (identical to the negative condition in Study 1): {Furthermore, taking antibiotics harms your gut microbiome. When you take antibiotics, they can kill the good bacteria in your gut. This, in turn, makes your gut more vulnerable to pathogens and has been linked to worse health outcomes for many different conditions.}

Societal: {Furthermore, taking unnecessary antibiotics contributes to the development of antibiotic resistance. When pathogens develop antibiotic resistance, antibiotics can be less effective for everyone. This means that people undergoing simple operations are more vulnerable to infectious complications, which can have fatal outcomes.}

#### 3.1.2. Statistical Analysis

Similar to Study 1, we conducted a three-way ANOVA for each outcome variable and, as stated in our pre-registration, used an SNK post hoc test to examine any significant main effects. The exploratory analyses tested whether the manipulation influenced the participants’ concerns about resistance, likelihood of seeking a second opinion, and likelihood of returning to the physician. The participants’ concerns about resistance were correlated with the primary and secondary outcomes. Linear models were used to test how their personal characteristics might affect their concerns about resistance.

### 3.2. Results

Although the participants in the societal frame were slightly less likely to continue to request antibiotics and more satisfied with their appointment, these differences were not significant. However, the participants who read the societal message were more likely to recommend the physician to a friend (*p* = 0.03, Table 4; and *p* = 0.009 for the ANOVA for a separate subset of the data of only societal vs. neutral).

In the exploratory analysis, we found that our manipulation had a significant effect on how concerned the participants were about antimicrobial resistance (*F* = 4.39, *p* = 0.01; societal vs. neutral *p* = 0.009). Furthermore, if a participant was more concerned about antibiotic resistance, they were less likely to ask for antibiotics (*r* = −0.26, *p* < 0.001) or seek a second opinion (*r* = −0.15, *p* < 0.001), and more likely to be satisfied with the appointment (*r* = 0.21, *p* < 0.001), return to the same doctor (*r* = 0.25, *p* < 0.001), and recommend the doctor (*r* = 0.29, *p* < 0.001). The manipulation had no significant effect on other secondary outcomes.

Overall, the participants were somewhat concerned about antimicrobial resistance (*M* = 3.27; on a scale from 1—not at all concerned to 5—very concerned). A third of the participants (*n* = 384, 36%) had previously had sinusitis and, of these participants, almost half (43%) reported previously receiving antibiotics for their sinusitis. Of the patients who had received antibiotics, 32% had been advised that antibiotics are not the best treatment compared to 58% who had not heard this. Of the patients who did not receive antibiotics for their sinusitis, 49% had heard this message, whereas 36% had not. A higher concern about resistance was observed among the participants who reported previously hearing that antibiotics are not the best treatment (*b* = −0.14, *p* = 0.03), older participants (*b* = 0.008, *p* = 0.02), participants with higher education (*b* = 0.17, *p* <0.001), and patients with better health (*b* = −0.08, *p* = 0.04). No difference was observed based on gender.

## 4. Discussion

These studies used a novel method to test the effect of various types of messages on participant expectations for antibiotics and expressed satisfaction with a medical appointment when antibiotics were not prescribed. The results suggest that a message about the societal effects of antibiotic resistance might have more impact, although its effect was only observed in one of the primary outcomes. However, the exploratory analyses showed that the societal message also had a stronger effect on participant concern about antibiotic resistance, which was further associated with all the primary and secondary outcomes. It also bears noting that in both studies any rationale (positive or negative frame, personal or societal frame) was slightly, non-significantly better than providing no additional rationale.

Discussing the effects of unnecessary antibiotics might not always appeal to doctors because of conflicts with patient expectations [11], time pressure, and the outcome not being a clear action [20]. Our results suggest, however, that providing messages about the effects of unnecessary antibiotics can affect participant perceptions about antibiotic resistance, and possibly exert a downstream effect on their expectations for antibiotics. Although multiple results from these studies point toward this conclusion, only one of the pre-registered primary outcomes show this effect. However, the results support previous research [12] in suggesting that patient perceptions are affected if physicians provide additional rationale about antibiotics and their risks. Conservatively, our results indicate that physicians taking the time to explain why they are not prescribing antibiotics does not hurt, and may even help patients understand and accept the recommended course of action.

We hypothesized that the personal message would be more effective than the societal one due to the importance of selfish motives [15]. Contrary to this hypothesis, the personal message was not more effective. In fact, if anything, the societal message was more effective (non-significantly for two variables and significantly for one). The societal message might have contained more novel information or seemed more impactful because it highlighted the potentially fatal consequences of antimicrobial resistance. Furthermore, the physician visit scenario was purely hypothetical, which might have made the selfish motive less urgent compared to a situation when a participant truly has symptoms (for a review of how hypothetical and real-world interventions can differ, see [21]). On the other hand, the situation may have reflected how people react to public messages about antibiotics [22]. A societal message including the possible severe consequences of unnecessary antibiotics might capture more attention and affect people’s concern about resistance.

This study has a number of limitations that should be considered. As mentioned, the situation was purely hypothetical and, therefore, caution is needed when generalizing the findings to real-life situations. Furthermore, this study’s population was somewhat skewed, such that older adults and people with lower levels of educational attainment were underrepresented. Additionally, this study was powered to detect a difference between the three conditions but not to examine differences among the subgroups based on, for example, age or previous experience of sinusitis.

Taken together, these studies demonstrate that randomized, controlled trials using online platforms and participant panels can be a tool with which to test various antibiotic messaging strategies before they are rolled out to physicians. Pre-testing interventions online is quick and cost-effective, and it reduces risks by testing hypothetical scenarios before their widescale implementation. By conducting power calculations, pre-registering the procedures and analyses, and randomizing the participants among the conditions, these studies provide a rigorous test of our hypotheses. Although the primary hypotheses are not consistently supported, the results demonstrate that the participants paid attention to details and showed the expected patterns of responses to the secondary variables; for example, the participants who reported being more concerned about antimicrobial resistant were less likely to request antibiotics. Thus, we think the results are valid and that further testing of this method for antibiotic messaging is warranted. In the case of positive vs. negative and personal vs. societal framing, it is possible that stronger messaging that better emphasizes the consequences of different choices would be more effective. Alternatively, different messaging strategies that draw on other behavioural insights (e.g., social norms) may be more effective. In addition, this method could be used to test various messages for situations where participants are not under the acute stress of a disease, such as public messaging on antibiotics. Lastly, randomized controlled trials testing the impacts of physicians’ antibiotic messaging strategies in real life should be explored to complement online testing.

These studies demonstrate that online, hypothetical studies are a viable method for testing messaging. Based on our results, discussing the consequences of different courses of action does not negatively impact physician–patient rapport. Thus, physicians may do well to discuss the effects of the use of unnecessary antibiotics with their patients. Notably, providing negative messages did not reduce how satisfied participants were with the physician, or whether they would recommend the physician to a friend. Indeed, the best-performing message was a negatively framed message that emphasized the societal consequences of unnecessary antibiotics. Therefore, these studies encourage physicians that it is okay to communicate the consequences of unnecessary antibiotics to their patients. These methods can also be applied to public health promotion more broadly. In fact, behavioural insights interventions and methodologies have been successfully used to tackle a number of public health challenges, such as vaccination [23], the obesity epidemic [24], and smoking cessation [25].

## 5. Conclusions

These studies demonstrate an inexpensive method with which to pre-test various messages advising patients against unnecessary antibiotic consumption, and suggest that such messages are not negatively received by patients.

## Figures and Tables

**Table 1 antibiotics-13-00657-t001:** Study 1 participant characteristics.

Age	Mean	35.4
	Range	19–81
Gender % (*n*)	Male	49% (523)
	Female	48% (513)
	Non-binary	2% (25)
	Prefer to self-describe	0.1% (1)
	Prefer not to answer	0.7% (7)
Education % (*n*)	Some schooling, but no diploma or degree	0.7% (7)
	High school diploma or GED	11% (116)
	Some college or post-secondary education	22% (231)
	College or post-secondary degree	46% (496)
	Some graduate school	4% (40)
	Graduate degree	17% (179)
Perceived health % (*n*)	Excellent	10% (105)
	Very good	41% (436)
	Good	36% (390)
	Fair	12% (127)
	Poor	1% (11)
Previous sinusitis	Yes	34% (365)
	No	50% (537)
	Unsure	16% (167)
Heard that antibiotics are not always needed for sinusitis	Yes	15% (159)
	No	71% (761)
	Unsure	14% (149)

**Table 2 antibiotics-13-00657-t002:** Average Likert-type scale ratings (standard deviations) for main outcomes across intervention groups and results of ANOVA for Study 1.

	Negative	Neutral	Positive	ANOVA
Request antibiotics	2.07 (1.07)	2.13 (1.11)	2.09 (1.10)	*F* = 0.23 (*p* = 0.79)
Appointment satisfaction	3.29 (0.98)	3.16 (1.04)	3.17 (1.00)	*F* = 1.75 (*p* = 0.18)
Recommend to a friend	3.28 (0.98)	3.20 (0.96)	3.25 (1.02)	*F* = 0.58 (*p* = 0.56)

**Table 3 antibiotics-13-00657-t003:** Study 2 participants’ characteristics.

Age	Mean	35.9
	Range	19–80
Gender % (*n*)	Male	50% (533)
	Female	48% (518)
	Non-binary	1% (16)
	Prefer to self-describe	0.1% (1)
	Prefer not to answer	0.5% (5)
Education % (*n*)	Some schooling, but no diploma or degree	0.7% (7)
	High school diploma or GED	11% (113)
	Some college or post-secondary education	21% (222)
	College or post-secondary degree	45% (485)
	Some graduate school	4% (44)
	Graduate degree	19% (202)
Perceived health % (*n*)	Excellent	12% (126)
	Very good	37% (392)
	Good	37% (401)
	Fair	12% (133)
	Poor	2% (21)
Previous sinusitis	Yes	36% (384)
	No	49% (523)
	Unsure	15% (166)
Heard that antibiotics are not always needed for sinusitis	Yes	18% (188)
	No	69% (736)
	Unsure	14% (149)

**Table 4 antibiotics-13-00657-t004:** Average Likert-type scale ratings (standard deviations) for main outcomes across various intervention groups and results of ANOVA for Study 2.

	Societal	Neutral	Personal	ANOVA
Request antibiotics	1.93 (1.10)	2.09 (1.10)	2.02 (1.08)	*F* = 2.05 (*p* = 0.13)
Appointment satisfaction	3.29 (1.03)	3.20 (1.02)	3.27 (1.01)	*F* = 0.68 (*p* = 0.51)
Recommend to a friend	3.40 (0.98)	3.22 (0.94)	3.33 (0.99)	*F* = 3.44 (*p* = 0.03); Societal vs. neutral difference 0.19, *p* = 0.03

## Data Availability

The data and the code are available on ResearchBox: https://researchbox.org/3256&PEER_REVIEW_passcode=MCDHXA (accessed on 10 July 2024).

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
