# Peer review of "Testing Different Message Styles about Unnecessary Antibiotics Using an Online Platform"

_antibiotics, 2024, doi:10.3390/antibiotics13070657_

Round 1

Reviewer 1 Report

Comments and Suggestions for Authors

Dear authors

Thank you for presenting this interesting article for publication. The effect of patient expectation on antibiotic use has long been debated and it is heartening to see research into addressing the problem. I have a few comments and/or queries

1. Lines 285 - 287: findings suggest that societal messaging has more of an impact than selfish messaging. Previous research has highlighted differences in knowledge and attitudes of various age groups towards antibiotic use. Did you examine each of your scenarios by age group and if so did you notice any differences? 

2. It is unclear if those who had previous experience of sinusitis responded any differently to the case study options to those who did not. 

3. minor typo: that abstract states 1,075 participants took part in S2 whoever Study 2 texts states 1,074.

Author Response

General notes: 

Thank you for reviewing our paper and for the supportive feedback. Please find below a point by point response to each reviewer’s comments. In addition, we wish to note that Reviewer 1 pointed out an inconsistency, which made us check the attention-check exclusions once more and realize that the attention check code had failed to run to the final analyses, resulting in a few more exclusions in Study 1. All the analyses in the paper were double-checked and corrected accordingly. In addition, while preparing and double-checking the entire code for publication on discrepancy in the exploratory analysis anova was detected and an inappropriate contingency table composition. All these have now been corrected and the code and data is available online. None of these small changes affected the interpretation of results. The data and analysis code are now available on Researchbox which will be made public upon publication: https://researchbox.org/3256&PEER_REVIEW_passcode=MCDHXA

Comment 1: Thank you for presenting this interesting article for publication. The effect of patient expectation on antibiotic use has long been debated and it is heartening to see research into addressing the problem. I have a few comments and/or queries
1. Lines 285 - 287: findings suggest that societal messaging has more of an impact than selfish messaging. Previous research has highlighted differences in knowledge and attitudes of various age groups towards antibiotic use. Did you examine each of your scenarios by age group and if so did you notice any differences? 

Response 1: 

Thank you for this interesting comment, which surely would be an interesting question for future research. This study was powered to detect a difference between the three intervention groups but not to explore subgroup differences. This is stated now as a limitation of the study on page 9 line 356.

“The study was powered to detect a difference between the three conditions but not to inform about differences in subgroups based on for example age or previous experience of sinusitis.”

Comment 2: It is unclear if those who had previous experience of sinusitis responded any differently to the case study options to those who did not. 

Response 2:

Thank you for this note. We further computed t-tests between experience of sinusitis and the measured variables. Below are the results of these analyses, but as results were inconsistent across the studies and did not provide clear insights we have not added them to the manuscript: 
Study 1: Patients who had previously had sinusitis were slightly more likely to ask for antibiotics (t = -2.56, p = .01), and be slightly less satisfied with the appointment (t = 2.09, p = .04). No significant difference by t-test was observed between previous sinusitis and seeking a second opinion (t = -0.58, p = .56), recommending the doctor (t = p.51, p = .60), or returning to the same physician (t = 0.10, p = .31).

Study 2: No significant correlation was observed between previous sinusitis and asking for antibiotics (t = -1.49, p = .14), satisfaction with the appointment (t = 1.07, p = .27), seeking a second opinion (t = 0.74, p = .46), recommending the doctor (t = 0.19, p = .85), or returning to the same physician (t = 0.81, p = .42). 

Comment 3. minor typo: that abstract states 1,075 participants took part in S2 whoever Study 2 texts states 1,074.

Response 3: Thank you for pointing this out. This prompted us to double-check and correct the participant and attention-check exclusions for both studies. Having corrected the attention check code, an additional 6 participants were excluded and the analysis estimates have been subsequently checked and amended. No change occurred that would affect the interpretation of the results.  

Reviewer 2 Report

Comments and Suggestions for Authors

Testing Different Message Styles on Unnecessary Antibiotics Using an Online Platform article by Stenlund et al. deals with the theme of antimicrobial resistance and patients' perception of this problem. It is quite interesting to to see the topic of AMR presented and investigated from this point of view- social, personal and psychological. The entire paper is performed and presented in a manner which reflects the diversity of research group. 

I have no specific comments nor requests for changes of the manuscript. Best of wishes in publishing your paper.

Author Response

General notes: Thank you for reviewing our paper and for the supportive feedback. Please find below a point by point response to each reviewer’s comments. In addition, we wish to note that Reviewer 1 pointed out an inconsistency, which made us check the attention-check exclusions once more and realize that the attention check code had failed to run to the final analyses, resulting in a few more exclusions in Study 1. All the analyses in the paper were double-checked and corrected accordingly. In addition, while preparing and double-checking the entire code for publication on discrepancy in the exploratory analysis anova was detected and an inappropriate contingency table composition. All these have now been corrected and the code and data is available online. None of these small changes affected the interpretation of results. The data and analysis code are now available on Researchbox which will be made public upon publication: https://researchbox.org/3256&PEER_REVIEW_passcode=MCDHXA

Comment 1: Testing Different Message Styles on Unnecessary Antibiotics Using an Online Platform article by Stenlund et al. deals with the theme of antimicrobial resistance and patients' perception of this problem. It is quite interesting to to see the topic of AMR presented and investigated from this point of view- social, personal and psychological. The entire paper is performed and presented in a manner which reflects the diversity of research group. 
I have no specific comments nor requests for changes of the manuscript. Best of wishes in publishing your paper.

Response 1: Thank you for positive feedback on our manuscript!

Reviewer 3 Report

Comments and Suggestions for Authors

As a health promotion lecturer, I find the paper interesting and relevant.

I have a few minor comments to improve the paper:

1) Introduction- some research about the public's lack of knowledge regarding antibiotic resistance and its reasons will enrich the introduction. See for example:  https://doi.org/10.3390/antibiotics12061028

2) Methods- The description of the sample characteristics should be presented in tables to make it more convenient for readers to understand. In general, presenting all findings in tables or graphs is recommended for better clarity.

3) Discussion- a section on the research limitations is missing. Please add.

4) Expand the conclusions section. You can explain how such findings can be utilized to promote public health, for example, to encourage vaccinations.

Author Response

General notes: Thank you for reviewing our paper and for the supportive feedback. Please find below a point by point response to each reviewer’s comments. In addition, we wish to note that Reviewer 1 pointed out an inconsistency, which made us check the attention-check exclusions once more and realize that the attention check code had failed to run to the final analyses, resulting in a few more exclusions in Study 1. All the analyses in the paper were double-checked and corrected accordingly. In addition, while preparing and double-checking the entire code for publication on discrepancy in the exploratory analysis anova was detected and an inappropriate contingency table composition. All these have now been corrected and the code and data is available online. None of these small changes affected the interpretation of results. The data and analysis code are now available on Researchbox which will be made public upon publication: https://researchbox.org/3256&PEER_REVIEW_passcode=MCDHXA

Comment 1: As a health promotion lecturer, I find the paper interesting and relevant.
I have a few minor comments to improve the paper:
Introduction- some research about the public's lack of knowledge regarding antibiotic resistance and its reasons will enrich the introduction. See for example:  https://doi.org/10.3390/antibiotics12061028

Response 1: Thank you for pointing this important point out. The following addition was made in the introduction on page 2, line 51: 

“Patients might not acknowledge the risks associated with unnecessary prescribing due to incomplete knowledge of antimicrobial resistance (Dopelt et al., 2023; Micallef et al., 2017).“

Comment 2: Methods- The description of the sample characteristics should be presented in tables to make it more convenient for readers to understand. In general, presenting all findings in tables or graphs is recommended for better clarity.

Response 2: Thank you for the suggestion. A Table has been added both for Study 1 and 2 to describe participants’ characteristics.

Comment 3: Discussion- a section on the research limitations is missing. Please add.

Response 3: Thank you for asking for this addition. We added a separate limitation section on page 9, line 352:  “The study has a number of limitations that should be considered. As mentioned, the situation was purely hypothetical and therefore caution is needed when generalizing the findings to real life situations. Furthermore, the study population is somewhat skewed, such that older adults and people with lower levels of educational attainment are underrepresented. Additionally, the study was powered to detect a difference between the three conditions but not to examine differences in subgroups based on, for example, age or previous experience of sinusitis.“

Comment 4: Expand the conclusions section. You can explain how such findings can be utilized to promote public health, for example, to encourage vaccinations.

Response 4: Thank you for this helpful suggestion. We’ve added a note discussing the broader applicability of our methods on page 10, line 386 although still kept the conclusion section concise and focused on our findings. 
“These methods can also be applied to public health promotion more broadly. In fact, behavioural insights interventions and methodologies have been successfully used to tackle a number of public health challenges, such as vaccination (see for example Milkman et al., 2024), the obesity epidemic (see for example Volpp et al., 2008), and smoking cessation (Volpp et al., 2006).”